# Early Prediction of Fetal Macrosomia Through Maternal Lipid Profiles

**DOI:** 10.3390/ijms26031149

**Published:** 2025-01-28

**Authors:** Vitaliy Chagovets, Natalia Frankevich, Natalia Starodubtseva, Alisa Tokareva, Elena Derbentseva, Sergey Yuryev, Anastasia Kutzenko, Gennady Sukhikh, Vladimir Frankevich

**Affiliations:** 1National Medical Research Center for Obstetrics, Gynecology and Perinatology Named After Academician Academician V.I. Kulakov of the Ministry of Healthcare of Russian Federation, 117997 Moscow, Russia; v_chagovets@oparina4.ru (V.C.); n_frankevich@oparina4.ru (N.F.); n_starodubtseva@oparina4.ru (N.S.); a_tokareva@oparina4.ru (A.T.); e_derbentseva@oparina4.ru (E.D.); g_sukhikh@oparina4.ru (G.S.); 2Moscow Center for Advanced Studies, 123592 Moscow, Russia; 3Department of Obstetrics and Gynecology, Siberian State Medical University, 634050 Tomsk, Russia; yurev.sy@ssmu.ru (S.Y.); kutzenko_aa@ssmu.ru (A.K.); 4Department of Obstetrics, Gynecology, Perinatology and Reproductology, Institute of Professional Education, Federal State Autonomous Educational Institution of Higher Education I.M. Sechenov, First Moscow State Medical University of the Ministry of Health of the Russian Federation (Sechenov University), 119991 Moscow, Russia; 5Laboratory of Translational Medicine, Siberian State Medical University, 634050 Tomsk, Russia

**Keywords:** fetal macrosomia, diagnostics, prognosis, lipidomics, mass spectrometry

## Abstract

The prevalence of fetal macrosomia is steadily increasing worldwide, reaching up to 20%. Fetal macrosomia complicates pregnancy and delivery. Current prediction strategies are inaccurate, and most patients with fetal macrosomia go into labor with an “unknown status”. The aim of this study was to develop a system for predicting fetal macrosomia based on the lipid profiles of pregnant women’s blood serum. In total, 110 patients were included in this study: 30 patients had gestational diabetes mellitus (GDM) and 80 did not. During the observation, blood samples were collected at three time points: in the first trimester (11–13 weeks of pregnancy), in the second trimester (24–26 weeks), and in the third trimester (30–32 weeks). Lipids were detected by flow injection analysis with mass spectrometry. Lipid profiles of pregnant women were discriminated by orthogonal projection on latent structure discriminant analysis (OPLS-DA) in all three trimesters. The developed OPLS-DA models allowed for the prediction of the occurrence of fetal macrosomia during pregnancy. Three sets of models were developed: models independent of GDM status with a sensitivity of 0.85 and specificity of 0.91, models for patients with positive GDM status with a sensitivity of 0.91 and specificity of 0.96, and models for patients with negative GDM status with a sensitivity of 0.93 and specificity of 0.92. Phosphatidylcholines and sphingomyelins were the most important discriminative features. These lipid groups probably play an important role in the pathogenesis of fetal macrosomia and may serve as laboratory markers of this pregnancy complication.

## 1. Introduction

The prevalence of metabolic disorders among pregnant women worldwide is steadily increasing, significantly varying based on factors such as dietary behaviors, cultural habits, average living standards in developed and developing countries, and maternal age. Annually, around 39 million pregnancies occur amidst maternal obesity, and, in some countries, its prevalence exceeds 60% [1]. Maternal overweight and obesity are independent risk factors for the development of gestational diabetes mellitus (GDM). The overall global standardized prevalence of GDM is 14.0% (95% CI: 13.97–14.04%). The prevalence varies across regions: North America and the Caribbean—7.1%, Europe—7.8%, South and Central America—10.4%, Africa—14.2%, the Western Pacific—14.7%, Southeast Asia—20.8%, and the Middle East and North Africa—27.6% [2]. As of 1 January 2019, in the Russian Federation, 4.58 million people (3.1% of the population) were registered with diabetes. Of these, 92% (4.2 million) had type 2 diabetes, 6% (256,000) had type 1 diabetes, and 2% (90,000) had other types of diabetes, including 8006 cases of gestational diabetes [3]. Fetal macrosomia is diagnosed three times more often in children born to women with GDM than in children born to mothers with normal blood glucose levels. However, a trend toward excessive fetal growth can also occur in women without carbohydrate metabolism disorders. Known non-modifiable risk factors include ethnicity, advanced maternal age (>35 years), multiparity, family history of diabetes or macrosomia, and male infants (*p* < 0.001) [4,5,6,7,8].

The risk of perinatal loss in fetal macrosomia is higher than in children with normal birth weight [9,10]. In pregnancies complicated by diabetes, fetuses with macrosomia develop a unique pattern of excessive growth characterized by central subcutaneous fat deposition around the abdomen and between the shoulder blades. This results in an increase in the shoulder circumference relative to the head, which, in 5–9% of cases, significantly increases the risk of shoulder dystocia, Erb’s palsy, brachial plexus injuries, fractures of tubular bones, and neonatal asphyxia [7,8,11]. For the mother, childbirth may be accompanied by complications such as prolonged labor and operative vaginal delivery or may result in a cesarean section. Postpartum complications may include trauma to the birth canal and uterine atony with an increased risk of postpartum hemorrhage [8,12]. Neonatal morbidity rates are higher among newborns with birth weights greater than 4000 g. Fetal macrosomia is associated with electrolyte and metabolic disturbances in the early neonatal period. For example, the incidence of severe hypoglycemia and neonatal hyperbilirubinemia is 5 and 2 times higher, respectively, compared to infants born to healthy mothers [7].

There are several theories regarding the pathogenesis of fetal macrosomia. The pathogenesis of fetal macrosomia is based on disturbances in carbohydrate and lipid metabolism within the “fetus–maternal” complex. These disruptions lead to excess glucose and lipids reaching the fetus, which contributes to excessive growth. Maternal hyperglycemia results in fetal hyperinsulinemia and an increase in fetal adipose tissue (Pedersen’s hypothesis) [7]. Changes in blood lipid profiles in pregnant women with gestational diabetes and macrosomia are currently a widely studied topic, though comprehensive and systematic data are yet to be established. A study on the serum lipid composition of healthy pregnant women in the first trimester showed that those with macrosomia had lower levels of lipids (phospholipids, lysophospholipids, monoacylglycerides) compared to women without macrosomia [13]. In mothers with metabolic disorders, the transplacental transfer of lipids is altered, which promotes accelerated differentiation of adipocytes in the fetus and leads to fetal macrosomia, excess weight, and obesity in children [14]. Transported fatty acids are thought to initiate the conversion of mesenchymal stem cells into adipocytes by activating transcription factors in the fetus. Excessive adipocyte formation leads to the development of excess weight. The source of these transferred lipids includes “free” fatty acids, triglyceride hydrolysis products, and components of the mother’s phospholipid polyunsaturated fatty acids [15,16,17]. Most of the fatty acids required by the fetus must be produced de novo, with approximately 20% being transferable through the placenta [18]. In pregnant women with diabetes, excess weight, and obesity, prolonged high plasma lipid concentrations (hyperlipidemia) result in excessive lipid transfer across the placenta. The fetus’s subcutaneous adipose tissue can synthesize fatty acids stimulated by insulin starting from the 10th week. The initial differentiation of adipocytes occurs between the 14th and 24th weeks of pregnancy [19,20]. During this period, the primary transplacental transfer of fatty acids takes place, and their increased intake is critical for the excessive formation of adipocytes in the fetus, leading to the birth of a large infant and the development of metabolic complications in the future. Thus, the exponential growth of adipose tissue in the fetus occurs simultaneously with the rise of lipids in the mother’s blood. However, the mechanisms behind the development of fetal macrosomia and its relationship with maternal hyperlipidemia constitute a complex, multifactorial process that requires further in-depth analysis. Additionally, high lipid levels in the mother’s blood correlate with maternal hypertension and preeclampsia [21,22].

A key challenge remains the development of early methods for predicting fetal macrosomia to reduce the occurrence of this complication both among pregnant women with metabolic disorders such as gestational diabetes and obesity and women without these risk factors.

The aim of this study was to develop a system for predicting fetal macrosomia based on the lipidomic profiles of pregnant women’s blood serum.

## 2. Results

### 2.1. Analysis of Clinical Characteristics

A comparative analysis of the clinical characteristics of the study groups was conducted (Appendix A). The groups were designated as GDM+ (the main group) and GDM− (the comparative group) for patients with and without GDM, respectively. The subgroups within each group were designated as Macrosomia+ and Macrosomia− for patients with and without macrosomia. The results of the analysis between the groups of women examined are presented in Appendix A. The analysis of the main and comparative groups did not reveal significant differences in the age and anthropometric data of the patients before pregnancy. The median age of the patients did not differ significantly between the groups and subgroups, with an average age of 32 years (*p* = 0.56). However, within-group analysis showed a difference between the Macrosomia+ and Macrosomia− subgroups. Pregnant women of the Macrosomia+ subgroups had a significantly higher pre-pregnancy body weight, especially those with GDM (84 (66;96), *p* = 0.006). Pre-pregnancy BMI significantly differed between the main GDM+ (22.6 (20.1; 26.5)) and comparative GDM− groups (21.2 (19.5; 22.9)) with *p* = 0.03). Patients with gestational diabetes and macrosomia started pregnancy with a significantly higher BMI compared to patients in other subgroups (27 (23; 30), *p* = 0.006). Weight gain by the time of delivery differed significantly among the patients with macrosomia, regardless of the presence of gestational diabetes (14 (11; 16) and 16 (13; 18), *p* = 0.006). Those with fetal macrosomia without gestational diabetes had the greatest weight gain by delivery. Additionally, women in this subgroup had a significantly higher birth weight (3.8 (3.5; 4.1), *p* = 0.004) compared to other patients. It is likely that genetic and behavioral factors like family diet and eating habits may equally contribute to the development of macrosomia in women without carbohydrate metabolism disorders. Patients in the GDM+ group were significantly more likely to undergo cesarean delivery compared to the GDM− group (18 (60%), 27 (34%), *p* = 0.02), reaching 71% in cases of fetal macrosomia. Cesarean deliveries in patients with gestational diabetes were mostly planned (13 (43%), 14 (18%), *p* = 0.01). Meanwhile, in patients with macrosomia without gestational diabetes, the nature of the cesarean delivery was mostly emergency (33%), which is known to be associated with more complications for both the mother and the fetus. The length of hospital stay after delivery varied considerably, with longer stays in the GDM+ group, regardless of fetal weight at birth. The terms were 5 (3; 5) for GDM+ and 4 (3; 5) for GDM− (*p* = 0.02). Neonatal outcomes including early neonatal complications, 1 and 5 min Apgar scores, and length of hospital stay did not differ significantly by GDM. However, when analyzing the Macrosomia+ and Macrosomia− subgroups, newborns with macrosomia remained in the hospital for significantly longer (4 (4; 5) and 4 (3; 5), *p* = 0.009).

Patients with macrosomia were significantly more likely to have a history of delivering a large baby, regardless of the presence of gestational diabetes (43% and 25% vs. 0% and 7%, *p* = 0.002). In addition, type 1 and type 2 diabetes in close relatives was more common in the Macrosomia+ subgroups for both the GDP+ (33%) and GDP− (57%) groups. A family history of diabetes among close relatives was found in 21% of cases in the GDM− group and 26% in the GDM+ group in the subgroups without macrosomia.

A comparative analysis was conducted on the frequency of excessive, insufficient, and recommended total weight gain during pregnancy among the women studied, using the criteria from the U.S. Institute of Medicine (2009). These criteria are based on the woman’s pre-pregnancy BMI to better differentiate between normal and pathological weight gain. The analysis showed that excessive weight gain during pregnancy was significantly more common among patients in the GDM+ group compared to those in the GDM− group (*p* < 0.001), while insufficient weight gain was significantly more frequent in the group of patients with fetal normosomia (*p* = 0.005). There were no significant differences in the frequency of recommended weight gain between the two groups (*p* = 0.16). Excessive weight gain in women with macrosomia also significantly exceeded that in patients with normal fetal weight at delivery (57% and 75% vs. 22% and 25%, *p* < 0.001).

Diagnoses of carbohydrate metabolism disorders were conducted in the first trimester of pregnancy using fasting venous blood glucose levels, in the second trimester using oral glucose tolerance test results, and in the third trimester based on fasting venous blood glucose levels. GDM was diagnosed in women with fetal macrosomia in 10.0% of cases in the first trimester, 70.0% in the second trimester, and 20.0% in the third trimester. Dynamic analysis of glycemia levels during the oral glucose tolerance test did not show significant postprandial glycemia in women with fetal macrosomia. GDM was diagnosed based on a single fasting venous blood glucose test in all patients with fetal macrosomia.

The primary therapeutic method for correcting hyperglycemia was dietary therapy, which involved excluding easily digestible carbohydrates (patients who required a switch to insulin therapy were excluded from further study). The recommended diet for gestational diabetes was followed by patients in 80.0% of cases. Adherence to dietary therapy was lower among patients with gestational diabetes who developed fetal macrosomia: only 50.0% complied with the diet.

### 2.2. Lipidomic Analysis of Blood Serum in Pregnant Women at 11–13, 24–26, and 30–32 Weeks of Pregnancy

A shotgun lipidomics analysis was conducted on the blood serum of 110 patients at 11–13, 24–26, and 30–32 weeks of pregnancy (330 samples in total). Using the OPLS-DA method, the samples were separated into Macrosomia− and Macrosomia+ subgroups based on lipid profiles at three stages of pregnancy: 11–13, 24–26, and 30–32 weeks (Figure 1).

The molecular profiles of blood serum could distinguish between samples from patients who later developed or did not develop fetal macrosomia.

The lipid profiles in the serum of patients with fetal macrosomia were characterized by increased levels of SM 32:7 and SM 33:5 and decreased levels of phosphatidylcholines, plasmalogen, and SM 34:1 (Figure 2).

In the next phase of this study, the potential of mass spectrometry to differentiate the blood serum of pregnant women with fetal macrosomia based on the presence or absence of GDM was evaluated. Lipidomic analyses of blood serum were conducted at 11–13 weeks, 24–26 weeks, and 30–32 weeks of pregnancy. OPLS-DA models were constructed for this purpose: one for patients with GDM (Figure 3) and one for patients without carbohydrate metabolism disorders (Figure 4).

The greatest differences in the blood lipidome between patients with and without macrosomia in the GDM+ group were found at 24 weeks of pregnancy, which coincided with the onset of GDM and the peak weight gain for most patients. For patients without GDM, the best separation of blood serum samples was observed at 11–13 weeks and 30–32 weeks of pregnancy.

At 11–13 weeks of pregnancy, the serum lipidome of patients with fetal macrosomia and GDM was characterized by elevated levels of PC 38:4, PC 38:3, and SM 32:7, along with a decrease in PC 36:5 (Figure 5a). At 24–26 weeks, an increase in LPC and PC levels was found in patients with macrosomia and GDM (Figure 5b). By 30–32 weeks, a decrease in PC and SM levels was observed in patients with macrosomia and GDM (Figure 5c).

Similarly, lipids associated with fetal macrosomia in the GDM− group were identified at three time points. At 11–13 weeks of pregnancy, levels of PC O-30:5, SM 32:7, SM 33:5, and SM 32:6 were elevated, while levels of SM 34:1, LPC 16:0, and several PCs were decreased in the macrosomia group (Figure 6a). At 24–26 weeks, an increase in LPC and SM levels along with a decrease in PC levels was observed in the macrosomia group (Figure 6b). At 30–32 weeks, levels of SM 32:7 and SM 33:5 were elevated, while levels of several PCs and SM 34:1 were decreased in the macrosomia group (Figure 6c).

The obtained data suggest that the serum lipid profiles of women with fetal macrosomia and those with normal fetal size showed significant differences throughout pregnancy (Table 1). Moreover, the predictive value of the lipid profile increased when identifying patients with GDM.

The characteristics of the developed OPLS-DA models that could be used to predict the occurrence of fetal macrosomia throughout pregnancy and the results of their ROC analyses are presented in Table 2. The Q^2^ values above 0.4 for the GDM+ group models indicated good predictive ability at all stages of pregnancy. Models based on serum lipid levels at 24 and 30 weeks of pregnancy, when fasting glucose data were unavailable, as well as those at 24 weeks in women without GDM were characterized by lower Q^2^ values. Figure 7 shows the ROC curves for each of the developed models along with the areas under the curves (AUCs). All AUC values were above 0.89.

## 3. Discussion

The main risk factors for macrosomia include maternal age, pre-pregnancy obesity, excessive weight gain before and during pregnancy, and GDM without insulin use [23,24,25,26,27]. In 2023, the Chinese Medical Association (JCMA) published data on maternal factors associated with fetal macrosomia in the Taiwanese population based on 4262 cases of full-term singleton births. According to the study, the significant risk factors identified were GDM, weight gain during the first six months of pregnancy (6 months GWG), and maternal BMI. The odds ratio (OR) for macrosomia was 3.1 in newborns of mothers with a 6 months GWG ≥ 15 kg, 6.3 for those born to mothers with GDM, and 4.1 for those born to mothers with a BMI ≥ 30 kg/m^2^, respectively. The authors emphasized the importance of counseling mothers to control weight both before and during pregnancy [23].

The analysis of our own clinical data yielded similar results in identifying significant risk factors for fetal macrosomia. These factors included higher pre-pregnancy body weight, with the highest values seen in patients with GDM; pre-pregnancy BMI; and total weight gain by the time of delivery (11–18 kg), regardless of GDM status. Notably, the greatest weight gain at delivery was observed in women with fetal macrosomia who did not have GDM. Additionally, it was interesting to find that women who delivered macrosomic babies had significantly higher birth weights themselves compared to other patients. Other factors included a history of delivering a large baby (regardless of GDM status) among multiparous women and a family history of type 1 or type 2 diabetes in close relatives (33–57%). These findings suggest that both genetic and behavioral factors, such as family diet and eating habits, may contribute equally to the development of macrosomia in women without carbohydrate metabolism disorders.

Despite extensive research on macrosomia, predicting which women are at risk remains challenging [27]. Antenatal risk factors are important for predicting macrosomia, but the outcome for both the fetus and the mother depends on the management of labor [24]. In our study, delivery by cesarean section reached 71% in cases of fetal macrosomia. Among patients with GDM, cesarean delivery was often planned, while, in cases of macrosomia without GDM, most cesarean deliveries (33%) were performed on an emergency basis, leading to a higher number of complications for both mother and baby. The length of hospital stay was significantly longer for patients with GDM, regardless of the infant’s birth weight. However, neonatal outcomes, such as early neonatal complications, Apgar scores at 1 and 5 min, and discharge time, did not differ significantly based on GDM status. The outcomes for newborns, however, were significantly different when accounting for birth weight. Infants with macrosomia had a notably longer hospital stay.

A retrospective cohort study by Dana Vitner and colleagues, which included 3098 mothers and children with macrosomia and spanned 15 years of observation (2000–2015), allowed for a comparison of management and outcomes between women with predicted fetal macrosomia and those whose fetal weight was unknown at the time of delivery. Primary outcomes included the frequency of cesarean sections (CSs) and postpartum hemorrhage, while secondary outcomes included combined maternal and neonatal outcomes and birth injuries. Macrosomia was predicted in 601 (19.4%) women, while, in 2497 (80.6%) cases, macrosomia was unknown. The rate of CS was more than 3.5 times higher in the predicted macrosomia group (47.2% vs. 12.7%, *p* < 0.001), consistent with our study results, where macrosomia was a predicted factor in GDM cases. The authors also noted a reduced risk of postpartum hemorrhage with an adjusted odds ratio (aOR) of 0.5 and a 95% confidence interval (95% CI) of 0.2–1.0 with planned CSs in the predicted macrosomia group, as well as reductions in other maternal complications (aOR 0.3, 95% CI 0.2–0.5) and adverse combined neonatal outcomes (aOR 0.7, 95% CI 0.6–0.9). Thus, the authors concluded that planned CSs, when macrosomia is predicted, lead to reduced risks of postpartum hemorrhage and improved maternal and neonatal outcomes, even for infants with a birth weight of less than 4500 g [28].

In cases where fetal macrosomia is suspected, patients should be thoroughly counseled about the delivery plan, and cesarean section should be considered when indicated. Methods for estimating fetal weight and predicting macrosomia include clinical measurements, ultrasound, and magnetic resonance imaging. However, current prediction strategies, such as clinical assessments and ultrasound, are inaccurate. Therefore, the search for new methods to predict and diagnose fetal macrosomia early is highly relevant today. The molecular mechanisms of dyslipidemia in fetal macrosomia remain largely unexplored, making research in this area highly promising.

In this study, we attempted to predict fetal macrosomia regardless of the presence or absence of GDM, starting as early as the first screening (11–13 weeks of pregnancy). Analysis of the lipid spectrum in the blood serum of women with fetal macrosomia and normal fetal weight revealed significant differences throughout pregnancy (at 11–13, 24–26, and 30–32 weeks). In a 2023 study by Yingdi Yuan et al., a predictive model for fetal macrosomia was developed based on maternal clinical and laboratory blood biomarkers that differed between women with GDM and macrosomia (GDM-M) and women with GDM and normal birth weight (GDM-N). The model included parameters such as pre-pregnancy BMI, weight gain by 24 weeks, parity, blood glucose levels two hours after an oral glucose tolerance test with 75 g of glucose at 24 weeks, HDL and LDL levels at 24 weeks, and the expression of CLUL1, VCAN, and RNASE3 in plasma at 24 weeks. This model showed good predictive efficiency for forecasting macrosomia in women with GDM [29]. Another 2023 study described comprehensive metabolite profiles in the serum of pregnant mothers and fetuses with normoglycemic macrosomia in a Chinese population. A total of 203 metabolites were identified, with lipids and lipid-like molecules predominating. Among them, 53 metabolites showed significant differences between samples of maternal venous blood and umbilical cord blood. These differences were observed in both the serum of pregnant women and in the fetuses with macrosomia [28]. In our study, lipid spectra in cases of fetal macrosomia showed significant differences throughout pregnancy in both patients with and without GDM while consistently highlighting the presence of phosphatidylcholines and sphingomyelins. It can be assumed that these lipid groups play a critical role in the pathogenesis of fetal macrosomia. Elevated levels of sphingomyelins may indicate decreased cellular sensitivity to insulin. Studies have found a positive correlation between sphingomyelin levels in adipocytes, insulin levels, and the insulin resistance index (HOMA-IR) in individuals with excess body weight [30,31]. An attempt to characterize and compare placental sphingolipid metabolism in type 1 diabetes (T1D), type 2 diabetes (T2D), and a control group without diabetes was conducted by Miira M. Klemetti and colleagues. Placental samples from T1D, T2D, and the control group were processed for sphingolipid analysis using tandem mass spectrometry. Western blotting, enzyme activity assays, and immunofluorescence were employed to study the enzymes regulating sphingolipids. The levels of ceramide in the placenta were found to be lower in T1D and T2D compared to the control group, which was associated with the increased expression of the enzyme that breaks down ceramide, acid ceramidase (ASAH1). Elevated ceramide levels in the placenta were observed in T1D complicated by preeclampsia. Similarly, higher ceramide levels were noted in pregnancies with poorly controlled glycemia in both T1D and T2D. The protein levels and activity of sphingosine kinase (SPHK), which produces sphingosine-1-phosphate (S1P), were highest in T2D. Additionally, SPHK levels were elevated in pregnancies with T1D and T2D associated with fetal macrosomia. In vitro experiments using JEG3 trophoblast cells demonstrated increased expression and activity of SPHK following glucose and insulin treatment. Specific alterations in the placental sphingolipids characterize placentas with T1D and T2D, depending on the type of diabetes and associated complications in both the fetus and mother. Increased insulin and glucose exposure is likely a contributing factor to the upregulation of the SPHK-S1P axis in placentas affected by diabetes [32]. In a study by Michal Ciborowski and colleagues, metabolic profiles of serum from healthy pregnant women were evaluated to identify early biomarkers of macrosomia and understand the mechanisms leading to abnormal fetal growth, independent of maternal body mass index or the presence of gestational diabetes. Lower levels of phospholipids, lysophospholipids, and monoacylglycerols; low metabolites of vitamin D3; and elevated bilirubin levels were associated with macrosomia. Since most of the changes were related to lipids, levels of the adipocyte fatty acid-binding protein (A-FABP) were measured as a validation concept, revealing a correlation with the studied lipids and birth weight. Serum fingerprinting in early pregnancy may predict the risk of macrosomia. Serum levels of A-FABP and several lipids are promising prognostic markers for macrosomia in healthy pregnancies [13]. According to a review published in 2021, a comprehensive summary of metabolomics studies in gestational diabetes (GDM) was conducted. The authors found that the pathways most commonly disrupted in GDM include amino acids (glutathione, alanine, valine, and serine), carbohydrates (2-hydroxybutyrate and 1,5-anhydroglycitol), and lipids (phosphatidylcholines and lysophosphatidylcholines). They also highlighted the potential use of certain metabolites as predictive markers for the development of GDM using highly stratified modeling methods [33].

Lysophosphatidylcholines are formed as a result of the partial hydrolysis of phosphatidylcholines by phospholipase A2. In their study, Patel N. et al. demonstrated a positive correlation between the levels of phosphatidylcholines and lysophosphatidylcholines in umbilical blood, the birth weight of infants born to obese mothers, and weight gain in children during the first six months of life. High levels of LPC 16:1 and 18:1 exhibited a linear relationship with hyperglycemia in women at 28 weeks of pregnancy [34]. In a study by Hellmuth et al. (2017), a positive correlation was found between newborn body weight and LPC 14:0, LPC 16:1, and LPC 18:1 in umbilical blood [35]. Another study demonstrated a positive correlation between LPC 14:0 and childhood obesity [36].

In our study, increased levels of LPC 16:0 were noted in the group of patients with fetal macrosomia, suggesting the potential prognostic significance of this lipid in this condition. Xiuli Su et al. applied untargeted metabolomic analysis to identify blood metabolites with high predictive potential for detecting type 2 diabetes (T2D) over a follow-up period of approximately 16 years. The metabolic profiles revealed significant disturbances in metabolomics even before the clinical onset and diagnosis of T2D. Overall metabolic shifts were closely related to insulin resistance rather than β-cell dysfunction. Additionally, 188 out of 578 annotated metabolites were associated with insulin resistance. Bidirectional mediational analysis revealed potential causal relationships between metabolites, insulin resistance, and the risk of T2D. Metabolomic analysis has potential clinical utility in predicting T2D [37].

The molecular mechanisms of dyslipidemia in fetal macrosomia remain poorly understood, and research in this area may not only enhance our understanding of the role of metabolic diseases but also contribute to the development of effective preventive measures aimed at reducing the incidence of this condition.

## 4. Materials and Methods

### 4.1. Study Design

A case–control study was conducted at the National Medical Research Center for Obstetrics, Gynecology, and Perinatology named after V.I. Kulakov in Moscow from January to September 2024 (Appendix A). Out of 1200 women who were monitored in the Scientific and Outpatient Department of the Kulakov Center starting from the first-trimester prenatal screening procedure (11–13.6 weeks), 110 patients were selected after delivery. Two groups were formed based on the presence (GDM+) or absence (GDM−) of gestational diabetes mellitus. The main GDM+ group included 30 patients, while the comparison GDM− group included 80. To address this study’s objectives, patients were further stratified into subgroups based on the presence (Macrosomia+) or absence (Macrosomia−) of fetal macrosomia. The Macrosomia+ subgroup of the GDM+ group included 7 patients with newborns weighing ≥ 4000 g and/or above the 90th percentile with GDM and the Macrosomia− subgroup included 23 patients with newborns weighing 2501 to 3999 g and GDM. The Macrosomia+ subgroup of the GDM-group included 24 patients with newborns weighing ≥ 4000 g and/or above the 90th percentile without GDM, while the Macrosomia− subgroup (control) included 56 patients with newborns weighing 2501 to 3999 g without GDM.

All patients signed voluntary informed consent to participate in this study. This work was approved by the Ethical Committee of the National Medical Research Center for Obstetrics, Gynecology, and Perinatology named after academician V.I. Kulakov (protocol no. 9, dated 22 November 2018).

A semi-quantitative assessment of serum lipid levels was performed using mass spectrometry.

Inclusion criteria for the main group were Caucasian race, singleton pregnancy, newborn weight between 2501 g and 4999 g with GDM, and patient consent to participate in this study.

Inclusion criteria for the comparison group were Caucasian race, absence of GDM, singleton pregnancy, newborn weight between 2501 g and 4999 g, and patient consent to participate in this study.

A mandatory condition for all patients’ inclusion in this study was participation in a comprehensive examination, which included three screening ultrasounds, venous blood collection, an oral glucose tolerance test at 24–28 weeks of pregnancy, and delivery at the center.

Exclusion criteria were type 1 and type 2 diabetes, any somatic pathology in the stage of decompensation, oncological diseases, autoimmune diseases, bronchial asthma in the stage of medical compensation, and multiple pregnancies.

Ultrasound examinations were conducted for all pregnant women at the designated times (11–14 weeks, 18–21 weeks, and 30–32 weeks of pregnancy). GDM was diagnosed using a glucose tolerance test with 75 g after 8–14 h of overnight fasting. After the first blood sample was taken, the plasma glucose level was measured within 30 min. If the glucose level exceeded 5.1 mmol/L after the first blood draw, the test was discontinued. If the test continued, the patient drank a glucose solution consisting of 75 g of dry glucose dissolved in 250–300 mL of warm (37–40 °C), non-carbonated water within 5 min. Subsequent blood draws for the determination of plasma glucose levels were performed 1 and 2 h after the glucose load. The threshold values for plasma glucose for diagnosing GDM and manifest diabetes are presented in Table 3 and Table 4.

### 4.2. Sample Collection

Blood samples for lipid analysis were collected at three points: the first at 11–13 weeks, the second at 24–26 weeks, and the third at 30–32 weeks of pregnancy. The samples were collected using a vacuum method into a sterile 9 mL S-Monovette tube containing a clot activator and separation granules, following a 12 h fasting diet. The serum was then centrifuged for 10 min at 700 g and 4 °C. The supernatant was carefully pipetted into a sterile tube, frozen, and stored at −80 °C. Blood was drawn from each patient at the specified times during this study: 11–13 weeks, 24–26 weeks, and 30–32 weeks of pregnancy.

### 4.3. Sample Preparation for Shotgun MS/MS

Lipid extracts were obtained using a modified Folch method. To 40 μL of serum, 480 μL of a chloroform–methanol mixture (2:1, *v*/*v*) was added. The mixture was incubated for 10 min and then filtered using filter paper, and 150 μL of a 1 mol/L NaCl aqueous solution was added to the resulting solution. The mixture was centrifuged at 3000 rpm for 5 min at room temperature. The organic lower layer containing the lipids was collected and dried under a nitrogen stream and then re-dissolved in a mixture of acetonitrile–isopropanol (1:1, *v*/*v*) for subsequent mass spectrometric analysis.

### 4.4. Shotgun MS Analysis of Serum Lipid Extracts

The molecular composition of serum samples was determined using flow injection analysis (FIA) electrospray ionization mass spectrometry with a Maxis Impact qTOF mass spectrometer (Bruker Daltonics, Bremen, Germany) [38,39]. Mass spectra were acquired over the *m*/*z* range of 400–1000 with the following parameters: capillary voltage at 4.1 kV, nebulizer gas pressure at 0.7 bar, drying gas flow rate at 6 L/min, and drying gas temperature at 200 °C. A constant flow of a methanol/water mixture (9:1, *v*/*v*) was supplied at a rate of 10 µL/min using a Dionex UltiMate 3000 binary pump (ThermoScientific, Bremen, Germany), and 20 µL of sample was injected via a Dionex UltiMate 3000 autosampler (ThermoScientific, Bremen, Germany).

Mass spectra were recorded in positive ion mode, achieving a resolution of 50,000 within the mass range of *m*/*z* 400–1000. For compound identification, tandem mass spectrometry (MS/MS) was performed in data-dependent mode. After a full mass scan, the five most abundant peaks were selected for MS/MS analysis, utilizing collision-induced dissociation with 35 eV of collision energy, a 1 Da isolation window, and a 1 min mass exclusion time.

After the mass spectrometric analysis, 100 mass spectra obtained during sample elution were averaged, normalized by total ion current (TIC), and converted into an abundance–*m*/*z* table for further processing. The spray remained stable throughout the analysis due to the constant flow and unchanged ion source parameters. The relative standard deviation of TIC during the integration time did not exceed 10%, while the intragroup relative standard deviation ranged from 10% to 20%.

### 4.5. Statistical Analysis

Statistical data processing was conducted using RStudio version 2023.06.1 with custom scripts written in R version 4.1.1. The Shapiro–Wilk test was employed to assess the normality of data distribution. For quantitative data that did not follow a normal distribution, median values (Me) and quartiles (Q1, Q3) were reported. Qualitative data were expressed as absolute values (%). Comparative analysis of qualitative data was performed using Fisher’s exact test and the Chi-square (χ^2^) test. For quantitative data, the Mann–Whitney test was used for pairwise comparisons between groups, while the Kruskal–Wallis test was applied for comparisons involving more than two groups. Bonferroni correction was utilized for multiple comparisons. The significance threshold was set at 0.05.

Mass spectrometry data analysis was carried out using multivariate analysis through Orthogonal Projections to Latent Structures Discriminant Analysis (OPLS-DA) [40], which allows for the construction of statistical models using multidimensional data to differentiate between samples. In building the linear regression model, the variable influence on projection (VIP) was calculated to assess the impact of individual X-variables (lipids) on the model. This helped to identify the most significant lipids and assess the statistical significance of differences in their levels between the study groups. Lipids were identified based on their accurate mass using the Lipid Maps database and by characteristic tandem mass spectra [41,42]. The lipid nomenclature corresponds to LipidMaps [42].

To determine the prognostic significance of the features, ROC analysis was performed, generating ROC curves and calculating the AUC.

## 5. Conclusions

Identifying at-risk groups for fetal macrosomia early in pregnancy allows healthcare providers to implement preventive measures focusing on lifestyle and dietary changes as early as the first trimester. This proactive approach can help tailor obstetric management to ensure the timely selection of the best delivery method, ultimately reducing adverse birth outcomes for both mothers and their babies.

Our research has identified clinical and experimental predictive markers for fetal macrosomia based on mass spectrometry analysis of blood samples. Key clinical risk factors include maternal age, pre-pregnancy obesity, excessive weight gain before and during pregnancy, and gestational diabetes (GDM) that has not progressed to insulin therapy. Our findings indicate that cesarean deliveries, often necessitated by fetal macrosomia, are typically performed in emergency situations, which increases the risk of complications for both mother and child. Additionally, newborn outcomes correlate strongly with their birth weight.

In this study, we aimed to predict fetal macrosomia regardless of whether GDM is present, starting as early as the first trimester. Our analysis of the lipid profiles in the serum of women with fetal macrosomia, compared to those with normal fetal weights, revealed significant differences throughout pregnancy (at 11–13 weeks, 24–26 weeks, and 30–32 weeks). We hypothesize that the lipid groups we identified, particularly phosphatidylcholines and sphingomyelins, play a critical role in the development of fetal macrosomia and could serve as laboratory markers for this complication.

We developed pilot OPLS-DA models that demonstrate high sensitivity and specificity for predicting fetal macrosomia during pregnancy. These models can be applied regardless of GDM status, making them useful for women with unknown GDM status as well as those with confirmed positive or negative diagnoses.

The molecular mechanisms that contribute to fetal macrosomia are largely uncharted, and further research in this area could enhance our understanding of the impact of metabolic diseases. This knowledge could lead to effective preventive strategies, improved prognostic techniques, and early diagnosis methods to help reduce the incidence of fetal macrosomia.

## Figures and Tables

**Figure 1 ijms-26-01149-f001:**
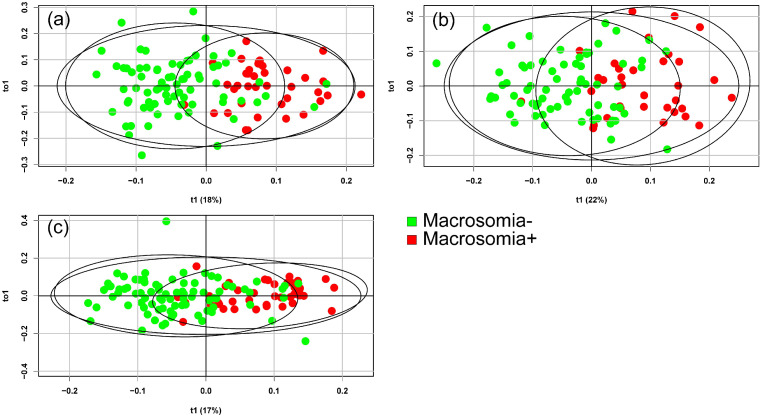
Score plots from the OPLS-DA analysis of mass spectrometry data discriminating between patients with and without macrosomia for all samples in the GDM+ and GDM− groups. Data for samples collected at (**a**) 11–13 weeks, (**b**) 24–26 weeks, and (**c**) 30–32 weeks of pregnancy. The component t1 on the x-axis is important for the segregation of the groups and maximizes the variance between the two groups. The orthogonal component (to1) on the y-axis is uncorrelated to t1 and maximizes the variance within the groups.

**Figure 2 ijms-26-01149-f002:**
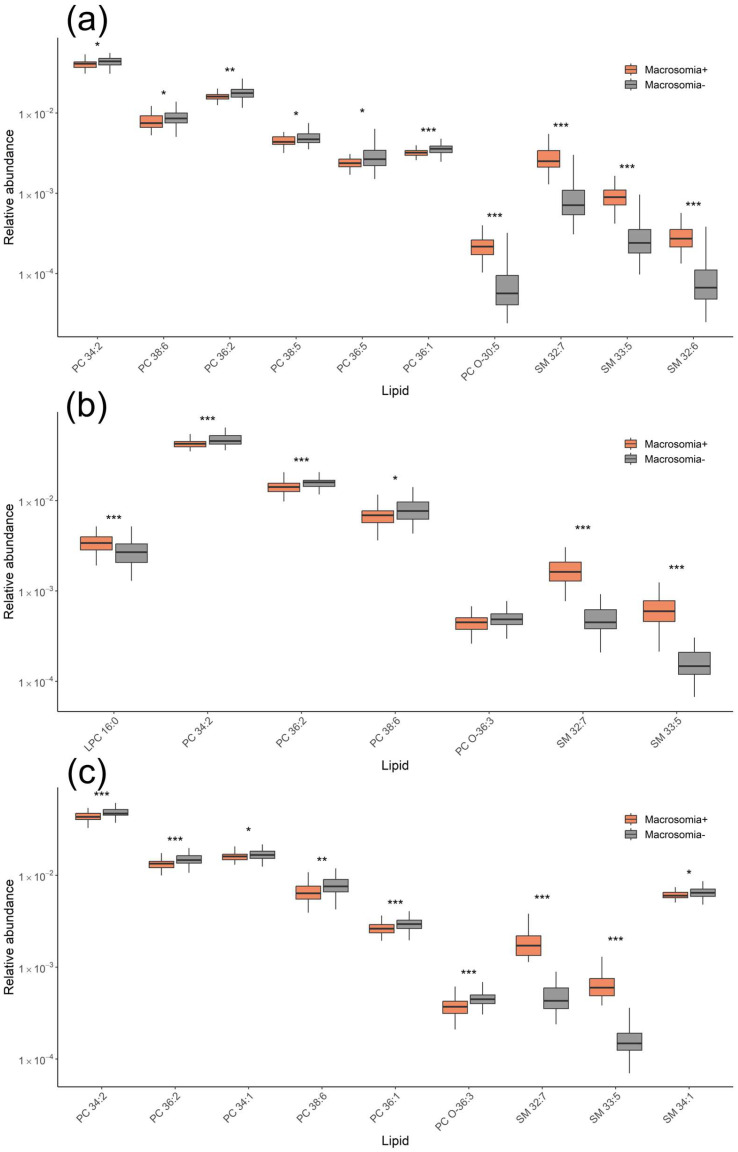
Lipid levels responsible for differentiating fetal macrosomia in both the GDM+ and GDM− groups for samples collected at (**a**) 11–13 weeks, (**b**) 24–26 weeks, and (**c**) 30–32 weeks of pregnancy. * *p* < 0.05; ** *p* < 0.01; *** *p* < 0.001.

**Figure 3 ijms-26-01149-f003:**
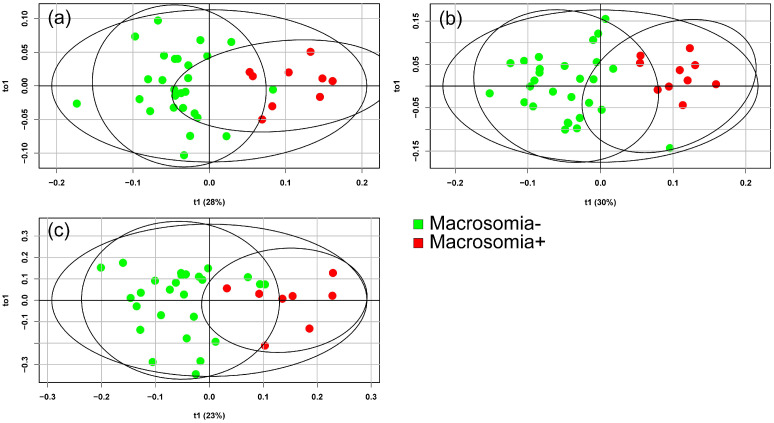
Score plots from the OPLS-DA analysis of mass spectrometry data discriminating between patients with and without macrosomia for samples in the GDM+ group. Data for samples collected at (**a**) 11–13 weeks, (**b**) 24–26 weeks, and (**c**) 30–32 weeks of pregnancy.

**Figure 4 ijms-26-01149-f004:**
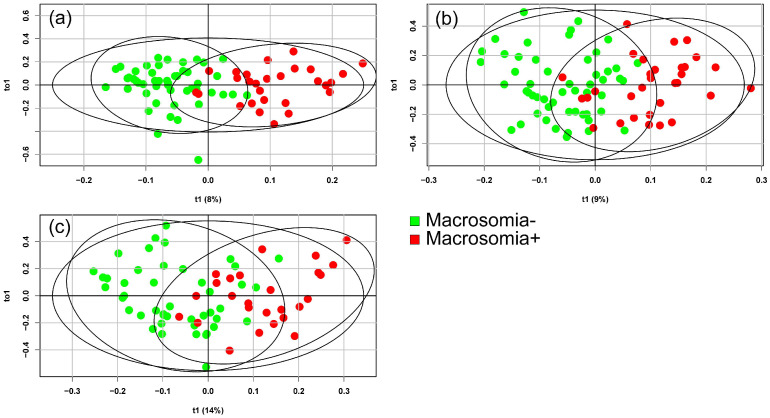
Score plots from the OPLS-DA analysis of mass spectrometry data discriminating between patients with and without macrosomia for samples in the GDM− group. Data for samples collected at (**a**) 11–13 weeks, (**b**) 24–26 weeks, and (**c**) 30–32 weeks of pregnancy.

**Figure 5 ijms-26-01149-f005:**
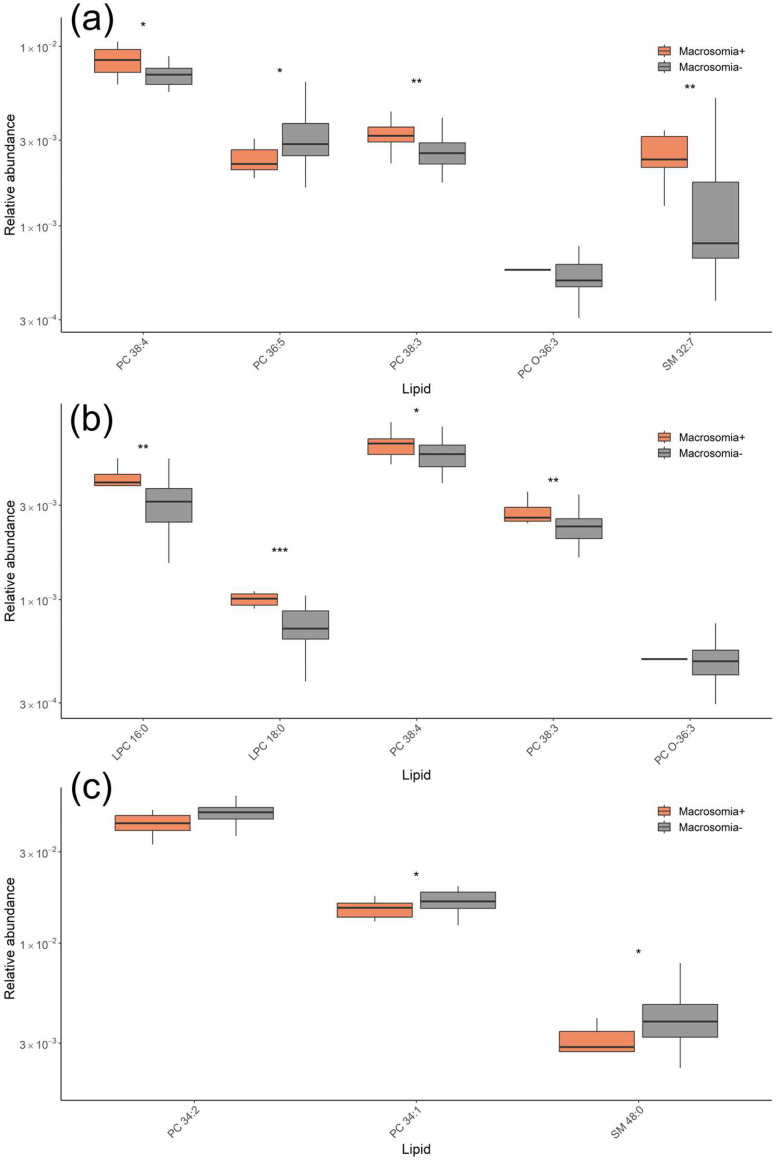
Lipid levels responsible for differentiating fetal macrosomia in the GDM+ group for samples collected at (**a**) 11–13 weeks, (**b**) 24–26 weeks, and (**c**) 30–32 weeks of pregnancy. * *p* < 0.05; ** *p* < 0.01; *** *p* < 0.001.

**Figure 6 ijms-26-01149-f006:**
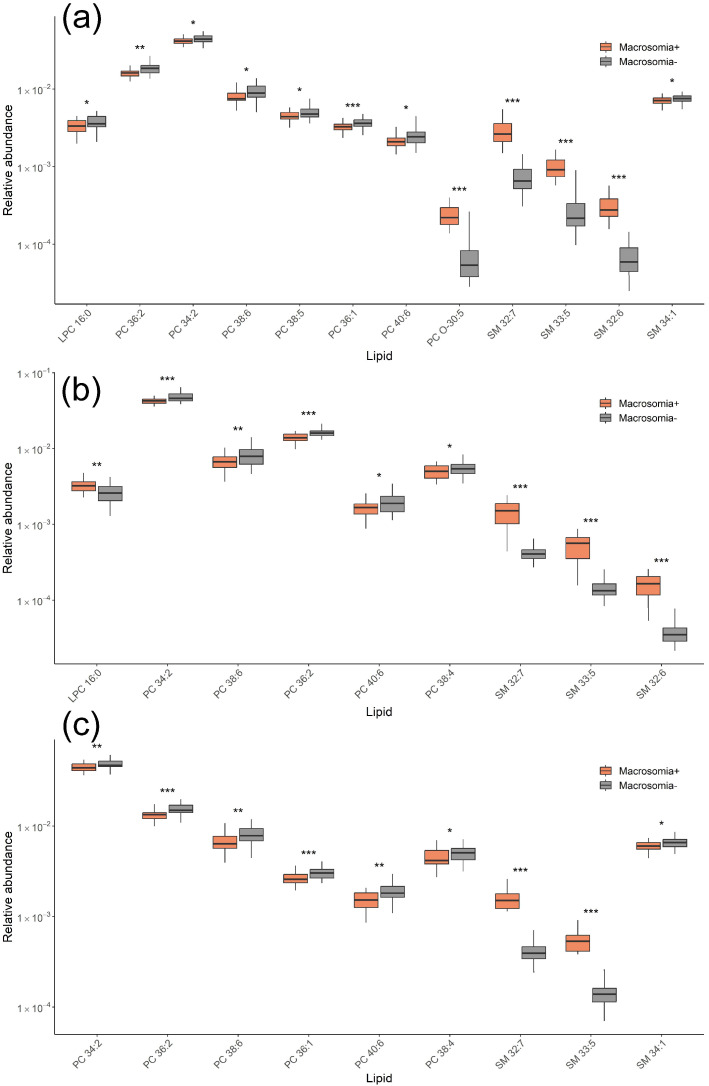
Lipid levels responsible for differentiating fetal macrosomia in the GDM− groups for samples collected at (**a**) 11–13 weeks, (**b**) 24–26 weeks, and (**c**) 30–32 weeks of pregnancy. * *p* < 0.05; ** *p* < 0.01; *** *p* < 0.001.

**Figure 7 ijms-26-01149-f007:**
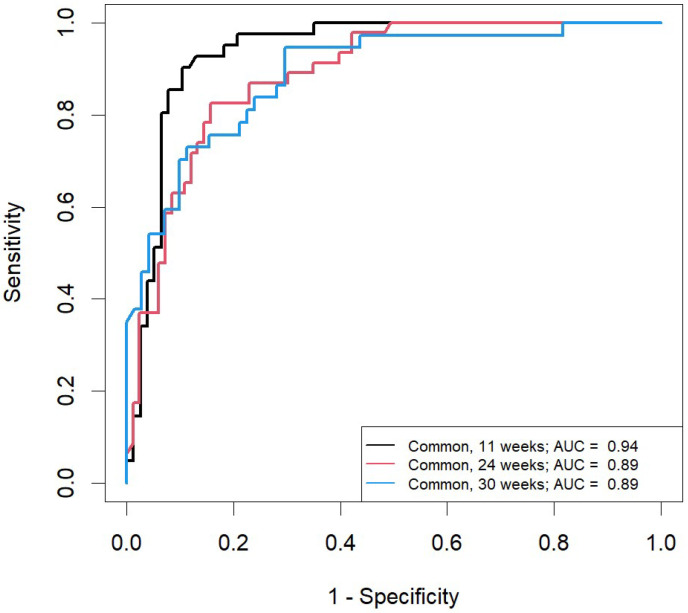
Results of ROC analyses of OPLS-DA models for predicting the development of fetal macrosomia.

**Table 1 ijms-26-01149-t001:** Lipids responsible for the differentiation of fetal macrosomia and normal fetal size depending on the presence or absence of GDM, estimated as being of variable importance in the projection according to the OPLS-DA analysis (VIP > 1).

GDM	Week of Pregnancy	Lipids with VIP > 1
No data	11–13	PC 34:2, PC 38:6, PC 36:2, PC 38:5, PC 36:5, PC 36:1, PC O-30:5, SM 32:7, SM 33:5, SM 32:6
	24–26	LPC 16:0, PC 34:2, PC 36:2, PC 38:6, PC O-36:3, SM 32:7, SM 33:5
	30–32	PC 34:2, PC 36:2, PC 34:1, PC 38:6, PC 36:1, PC O-36:3, SM 32:7, SM 33:5, SM 34:1
Confirmed	11–13	PC 38:4, PC 36:5, PC 38:3, PC O-36:3, SM 32:7
	24–26	LPC 16:0, LPC 18:0, PC 38:4, PC 38:3, PC O-36:3
	30–32	PC 34:2, PC 34:1, SM 48:0
Excluded	11–13	LPC 16:0, PC 36:2, PC 34:2, PC 38:6, PC 38:5, PC 36:1, PC 40:6, PC O-30:5, SM 32:7, SM 33:5, SM 32:6, SM 34:1
	24–26	LPC 16:0, PC 34:2, PC 38:6, PC 36:2, PC 40:6, PC 38:4, SM 32:7, SM 33:5, SM 32:6
	30–32	PC 34:2, PC 36:2, PC 38:6, PC 36:1, PC 40:6, PC 38:4, SM 32:7, SM 33:5, SM 34:1

**Table 2 ijms-26-01149-t002:** Characteristics of the OPLS-DA models for predicting fetal macrosomia at 11–13 weeks, 24–26 weeks, and 30–32 weeks of pregnancy. Rows in bold correspond to models with good predictive ability (Q^2^ > 0.4).

GDM	Week of Pregnancy	R^2^X	R^2^Y	Q^2^	Number VIP > 1	AUC	Sensitivity	Specificity
No data	**11–13**	**0.480**	**0.509**	**0.418**	**20**	**0.94**	**0.85**	**0.91**
	24–26	0.430	0.419	0.276	25	0.89	0.67	0.88
	30–32	0.472	0.398	0.301	20	0.89	0.59	0.90
Confirmed	**11–13**	**0.557**	**0.612**	**0.428**	**13**	**0.94**	**0.91**	**0.96**
	**24–26**	**0.614**	**0.643**	**0.487**	**19**	**0.95**	**0.93**	**0.96**
	**30–32**	**0.717**	**0.603**	**0.449**	**12**	**0.97**	**0.90**	**0.88**
Excluded	**11–13**	**0.470**	**0.674**	**0.601**	**21**	**0.98**	**0.93**	**0.92**
	24–26	0.446	0.629	0.231	23	0.95	0.75	0.98
	**30–32**	**0.508**	**0.601**	**0.496**	**22**	**0.94**	**0.85**	**0.93**

**Table 3 ijms-26-01149-t003:** Threshold values for diagnosing GDM during the oral glucose tolerance test with 75 g of glucose.

Venous Plasma Glucose	mmol/L	mg/dL
Fasting venous plasma glucose	≥5.1	≥92
After 1 h	≥10.0	≥180
After 2 h	≥8.5	≥153

**Table 4 ijms-26-01149-t004:** Threshold values for diagnosing manifest diabetes mellitus in pregnant women.

Fasting Venous Plasma Glucose	≥7.0 mmol/L	≥126 mg/dL
Glycated hemoglobin (HbA1c)	≥6.5%	
Venous plasma glucose regardless of food intake	≥11.1 mmol/L	≥200 mg/dL

## Data Availability

Data are contained within the Appendix A.

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
