# Peer review of "Early Prediction of Fetal Macrosomia Through Maternal Lipid Profiles"

_ijms, 2025, doi:10.3390/ijms26031149_

Round 1
Reviewer 1 Report
Comments and Suggestions for Authors
A very interesting approach to the important problem of fetal macrosomia and its' consequences. It would be of great interest to conduct a prospective cohort study in larger groups of patients.
I would suggest restructuring the manuscript to fit in the classic rules of constructing a manuscript. Some references are too old and should be exchanged for more recent ones.
Author Response
A very interesting approach to the important problem of fetal macrosomia and its' consequences. It would be of great interest to conduct a prospective cohort study in larger groups of patients.
I would suggest restructuring the manuscript to fit in the classic rules of constructing a manuscript. Some references are too old and should be exchanged for more recent ones.
Answer: Thank you for your high evaluation of our work. The structure of the manuscript sections is determined by the journal's editorial requirements. The references have been updated to more recent ones relevant to the topic at hand.
Reviewer 2 Report
Comments and Suggestions for Authors
This is an interesting paper on shot gun mass sectrometric analysis in women throughout pregnancy, aiming at the definition of predictive factors. However, there are severy major and e few minor comments to be made.
Major:
The manuscript is too wordy and narrative, and must be shortened and written more focused.
Many parts in the Results section belong in a clear-cut form to the Methods (l.108-112, 160-163, 168-170, 176-179, ) that should be placed, as mostly usual, between the Introduction and Results sections.
Generally, a flow sheet (Fig. 1) starting with all ! patients over recrutation, exclusion reasons and numbers until inclusion must be provided. How was pregnancy verified? Weren't there any drop outs? If yes, were all their data excluded? Is the number of patients representative for the population, as many women do not know at the first study time point that they are pregnant. L. 434: The text is unclear, as it means that only patients with GDM are included.
L. 412ff: Provide a table. Write out the group descriptions rather than using a minus symbol.
Discussion: While broadly describing other studies and their findings, it is lengthy and sub-headings are missing. The reviewer misses a positioning of the authors’ data in relation to these, and what the advantage of tandem mass spectrometry is. The comments on our lack of our understanding are trivial and should be left away or specified.
Fig. 3+4: There is no indication, how the score is defined and what relative abundance means.
All Figures:The fonts/letter sizes are too small throughout. They must be as larges as the fonts in the manuscript.The abbreviations used are not explaned, even not in the first figure legend. Notably, whereas this terminology is chemically correct, it is undigestible or even frightening to clinicians, and nowhere in the manuscript the difference of a diacyl- (PC) and alkyl-acyl-PC (PC-O) or the meaning of PC38:6, apparently 1-palmitoy-2-docosahexaenoyl-glycerophosphocholine, is digestibly described. Moreover, in the discussion, any reference to these poly-unsaturated phospholipids is missing.
Finally, the authors did not determine absolute concentrations using internal standards (like odd-numbered or stable isotope-labeled PCs). Hence, any change in the maternal SMs and PC-Os being associated with fetal macrosomia cannot be related to concentrations, and it remains elusive, whether these componds are increase in macrosomia, or whether the sum of other compounds is decreased.
Table 1: Define abbreviation VIP>1. Restructure table to a more convenient design, e.g. with black horizontal lines and with differentiating phospholipid groups (LPC, PC, PC-O, SM in separate lines.
L. 247 and elsewhere: The description of the OPLS-DA models includes what? It's not easy to get the point in connecting the data with the model as the Methods section is a bit poor.
What is missing are data on total concentrations by using an internal standard. Hence, it remains unclear, whether increases in macroosomia+ patients are are due to a real increase or a relative increase due to a decrease of others.
L. 37, 51 and elsewhere: The major influence of nutrition rather than simply age and and ethnicity must be highlighted. It may be more related to the increase in overweight with increasing age, and overfeeding in so-called 'developed' countries.
L. 85-87:Most of the fatty acids required by the fetus must be produced de novo, with approximately 20% being transferable through the placenta.
L. 85-86 Synthesized by whom? And this doesn't apply to ARA and DHA being conditionally essential to the fetus. Please specify!
L. 345-346: I do not agree that an association of different levels means causality.
Minor:
L38: 'some countries': Mention examples for either condition.
L. 42-44: Why is a minus in front of the figures? Explain and change, please.
L.47: Relate the mumbers of cases to the numbers of pregancies, please.
L. 53: in fetal, inchildren: correct for othography
L. 78f: weird wording, as transplacental lipid transfer occurs anyway! Please reword.
Fig 7: The green line nearly invisible. Use eiter one colour or solid lines throghout.
Line 229: Superfluous words; 'were identified at three time points'.
L. 269: >this< rather than >the<,
L. 281f: Provide data or Ref.
L. 287-299: Refs and/or data are missing.
L. 341f: >metabolic disturbances<... Define these disturbances.
L.345: Well, these are always present. Do the authors mean concentrations/'levels'/fractions of characteristic PCs and SPMs? Then give them names, please.
L357: Do the authors mean expression rather than 'regulation'?
L 375: 'was' rather than 'were'.
L. 418: Define, what main and comparison GDM group means.
L. 424: 'groupincluded' is two words.
L. 428ff: Provide name of institution only, date, and full code name.
L. 537: Please differentially mention PCs, PC-Os and SPMs.
Author Response
This is an interesting paper on shot gun mass sectrometric analysis in women throughout pregnancy, aiming at the definition of predictive factors. However, there are severy major and e few minor comments to be made.
Major:
The manuscript is too wordy and narrative, and must be shortened and written more focused.
Many parts in the Results section belong in a clear-cut form to the Methods (l.108-112, 160-163, 168-170, 176-179, ) that should be placed, as mostly usual, between the Introduction and Results sections.
Answer: We agree with the respected reviewer; however, the structure of the manuscript is dictated by the journal's requirements (the Methods section must be placed at the end of the manuscript, following the Discussion). The four sections you highlighted in the Results that describe the methodology were intentionally included by us to enhance the reader's understanding, as it is inconvenient to refer back to the Materials and Methods section to clarify particularly important steps that are first described in the Results section. However, we believe that these transitions do not overcrowd the manuscript and we would very much like to retain them, as they add literary value and facilitate comprehension.
Generally, a flow sheet (Fig. 1) starting with all ! patients over recrutation, exclusion reasons and numbers until inclusion must be provided. How was pregnancy verified? Weren't there any drop outs? If yes, were all their data excluded? Is the number of patients representative for the population, as many women do not know at the first study time point that they are pregnant. L. 434: The text is unclear, as it means that only patients with GDM are included.
Answer: The Figure S1 is now provided in the manuscript. All data from the patients who were excluded from the study (n=1090) were not used in the subsequent analysis. At the time of inclusion in the study (11-13.6 weeks of pregnancy), all patients were aware of their pregnancy and sought care at the Center for their first screening. None of the patients had a history of alcohol or drug dependence that could have affected the course of the pregnancy during the very early stages (before the patients were aware of their pregnancy). The confirmation of pregnancy was based on ultrasound findings and blood tests, specifically the increase in the beta subunit of human chorionic gonadotropin (ranging from 10 to 300,000 mIU/mL, peaking at 8-11 weeks of gestation)
- 412ff: Provide a table. Write out the group descriptions rather than using a minus symbol.
Answer: The table is provided.
Discussion: While broadly describing other studies and their findings, it is lengthy and sub-headings are missing. The reviewer misses a positioning of the authors’ data in relation to these, and what the advantage of tandem mass spectrometry is. The comments on our lack of our understanding are trivial and should be left away or specified.
Answer: To improve the style of the article, we removed a few sentences from the text (L. 263-265).
Fig. 3+4: There is no indication, how the score is defined and what relative abundance means.
Answer: The score plot is one of the standard representations of the OPLS-DA analysis result. For more information, see reference [39]. Relative abundance refers to the intensity normalized by total ion current, which is described in the Materials and Methods section.
All Figures:The fonts/letter sizes are too small throughout. They must be as larges as the fonts in the manuscript.
Answer: The fonts in the figures have been enlarged.
The abbreviations used are not explaned, even not in the first figure legend. Notably, whereas this terminology is chemically correct, it is undigestible or even frightening to clinicians, and nowhere in the manuscript the difference of a diacyl- (PC) and alkyl-acyl-PC (PC-O) or the meaning of PC38:6, apparently 1-palmitoy-2-docosahexaenoyl-glycerophosphocholine, is digestibly described.
Answer: The lipid nomenclature corresponds to LipidMaps [41]. The reference has been added to the Materials and Methods section.
Moreover, in the discussion, any reference to these poly-unsaturated phospholipids is missing.
Answer: There are no clear data in the literature on the role of individual lipids species, so in this study they are used as biomarkers to identify the current state of the patient and predict the direction of its change.
Finally, the authors did not determine absolute concentrations using internal standards (like odd-numbered or stable isotope-labeled PCs). Hence, any change in the maternal SMs and PC-Os being associated with fetal macrosomia cannot be related to concentrations, and it remains elusive, whether these componds are increase in macrosomia, or whether the sum of other compounds is decreased.
Answer: The present study focused on the overall lipid profile rather than the absolute concentration of some specific lipids therefore normalization by total ion current was used.
Table 1: Define abbreviation VIP>1. Restructure table to a more convenient design, e.g. with black horizontal lines and with differentiating phospholipid groups (LPC, PC, PC-O, SM in separate lines.
Answer: An explanation of the VIP abbreviation has been added to the table header. The focus of the table is on patient groups rather than lipid classes, so it would be inconvenient to separate lipid classes with horizontal lines.
- 247 and elsewhere: The description of the OPLS-DA models includes what? It's not easy to get the point in connecting the data with the model as the Methods section is a bit poor.
What is missing are data on total concentrations by using an internal standard. Hence, it remains unclear, whether increases in macroosomia+ patients are are due to a real increase or a relative increase due to a decrease of others.
Answer: The present study focused on the overall lipid profile rather than the absolute concentration of some specific lipids therefore normalization by total ion current was used. Since the total ion current sums the intensities of several hundred ions, a statistically significant difference in the relative intensity of a single ion between groups is more likely to be due to a difference in the level of the corresponding compound.
- 37, 51 and elsewhere: The major influence of nutrition rather than simply age and and ethnicity must be highlighted. It may be more related to the increase in overweight with increasing age, and overfeeding in so-called 'developed' countries.
Answer: the text was changed to: The prevalence of metabolic disorders among pregnant women worldwide is steadily increasing, significantly varying based on factors such as dietary behaviors, cultural habits, average living standards in developed and developing countries, and maternal age
- 85-87:Most of the fatty acids required by the fetus must be produced de novo, with approximately 20% being transferable through the placenta.
- 85-86 Synthesized by whom? And this doesn't apply to ARA and DHA being conditionally essential to the fetus. Please specify!
Answer: In pregnant women with diabetes, overweight, and obesity, the prolonged high concentration of lipids in plasma (hyperlipidemia) leads to an excessive transfer of lipids across the placenta. From 14 to 24 weeks of gestation, the primary transplacental transfer of fatty acids occurs, and their increased influx is critical for the excessive formation of adipocytes in the fetus, resulting in larger birth sizes and the potential for future metabolic complications. Thus, the exponential growth of adipose tissue in the fetus occurs simultaneously with the increase of lipids in the mother’s blood.
However, it should be noted that neither phospholipids nor triglycerides directly cross from maternal blood to the fetus. The intra-placental transport and metabolism of fatty acids occur through several pathways, either by using intact molecules or after prior modification through desaturation, elongation, or partial oxidation. Intermediate steps of esterification may occur to form phospholipids and triglycerides, which are temporarily stored before being hydrolyzed by phospholipases and acylglycerol lipases to release fatty acids. Additionally, de novo synthesis can serve as another source of fatty acids in the placenta [Baschat A.A. Fetal responses to placental insufficiency: an update // BJOG. – 2004. – Vol. 111, No. 10, – P. 1031–1041.2]. The reference has been added to the Literature section.
- 345-346: I do not agree that an association of different levels means causality.
Answer: Dear Reviewer, we certainly understand your concerns. However, we drew this conclusion based on our observations within the framework of our hypothesis. Further research involving a larger patient sample will allow us to validate this assumption.
Minor:
L38: 'some countries': Mention examples for either condition.
Answer: Lines 39-47 include more detailed information on the prevalence of the studied issue in various countries around the world.
- 42-44: Why is a minus in front of the figures? Explain and change, please.
Answer: Hyphens have been removed from the text of the article.
L.47: Relate the mumbers of cases to the numbers of pregancies, please.
Answer: We utilized officially published data from the Russian Association of Endocrinologists and the Russian Society of Obstetricians and Gynecologists on Gestational Diabetes Mellitus: Diagnosis, Treatment, Obstetric Tactics, and Postpartum Monitoring. Clinical Guidelines, 2020 (link attached). Population studies analyzing the epidemiological situation regarding gestational diabetes did not fall within the scope and objectives of our research.
- 53: in fetal, inchildren: correct for othography
Answer: Corrected
- 78f: weird wording, as transplacental lipid transfer occurs anyway! Please reword.
Л. 78f: странная формулировка, так как трансплацентарный перенос липидов происходит в любом случае! Пожалуйста, перефразируйте.
Answer: The text was changed to: In mothers with metabolic disorders, the transplacental transfer of lipids is altered, which promotes accelerated differentiation of adipocytes in the fetus and leads to fetal macrosomia, excess weight, and obesity in children.
Fig 7: The green line nearly invisible. Use eiter one colour or solid lines throghout.
Answer: Corrected
Line 229: Superfluous words; 'were identified at three time points'.
Answer: The phrase is used for clarification only.
- 269: >this< rather than >the<,
Answer: Corrected
- 281f: Provide data or Ref.
- 287-299: Refs and/or data are missing.
Answer: These are the data from our study. They are presented in detail in the Results section and the clinical table (see Appendix).
- 341f: >metabolic disturbances<... Define these disturbances.
Answer: The text was changed to: Among them, 53 metabolites showed significant differences between samples of maternal venous blood and umbilical cord blood. These differences were observed in both the serum of pregnant women and in the fetuses with macrosomia. [28]
L.345: Well, these are always present. Do the authors mean concentrations/'levels'/fractions of characteristic PCs and SPMs? Then give them names, please.
Answer: Correceted
L357: Do the authors mean expression rather than 'regulation'?
Answer: Corrected.
L 375: 'was' rather than 'were'.
Answer: In this sentence ‘were’ is related to ‘levels’.
- 418: Define, what main and comparison GDM group means.
Answer: The table (Figure S1) was added to the text.
- 424: 'groupincluded' is two words.
Answer: It has been corrected.
- 428ff: Provide name of institution only, date, and full code name.
Answer: Corrected
- 537: Please differentially mention PCs, PC-Os and SPMs.
Answer: PC-Os are ment here as a subclass of phosphatidylcholines.
Reviewer 3 Report
Comments and Suggestions for Authors
This paper presents a study of a method to use lipid measurements in pregnant woman to improve prediction of the risk of fetal macrosomia both in women with gestational diabetes (GDM) and those without GDM.
This is a valuable paper, but the presentation of the material needs to be improved.
MAJOR ISSUES
The material is presented in an unusual order. The methods are presented between the discussion and the conclusion. I strongly suggest that the methods are placed between the introduction and the results section, as is conventional.
The material on pages 11 to 12 as part of the discussion would be better as part of the introduction / background.
The conclusion should make clear to what extent the method presented here is likely to offer an improvement over existing methods of predicting fetal macrosomia during pregnancy.
MINOR POINTS
Lines 136 to 138
“The length of hospital stay after delivery varied considerably, with longer stays in the GDM+ group, regardless of fetal weight at birth.. The terms were 5(3;5) for GDM+ and 4(3;5) for GDM- (p = 0.02).”
Please make clear what units are being used for the hospital stays.
Figure 1 (and other similar figures)
It’s not clear to me what is being plotted on the x and y axes.
Author Response
This paper presents a study of a method to use lipid measurements in pregnant woman to improve prediction of the risk of fetal macrosomia both in women with gestational diabetes (GDM) and those without GDM.
This is a valuable paper, but the presentation of the material needs to be improved.
MAJOR ISSUES
The material is presented in an unusual order. The methods are presented between the discussion and the conclusion. I strongly suggest that the methods are placed between the introduction and the results section, as is conventional.
The material on pages 11 to 12 as part of the discussion would be better as part of the introduction / background.
Answer: Thank you for your high evaluation of our work. The structure of the manuscript sections is determined by the journal's editorial requirements. The entire text of the article has been significantly revised accordance with the comments.
The conclusion should make clear to what extent the method presented here is likely to offer an improvement over existing methods of predicting fetal macrosomia during pregnancy.
Answer: the Conclusion section has been rewritten.
MINOR POINTS
Lines 136 to 138
“The length of hospital stay after delivery varied considerably, with longer stays in the GDM+ group, regardless of fetal weight at birth.. The terms were 5(3;5) for GDM+ and 4(3;5) for GDM- (p = 0.02).”
Please make clear what units are being used for the hospital stays.
Answer: The term "bed-day" was used. A bed-day is an economic unit of measure equal to one day of occupancy for a person occupying a sleep space (bed).
Figure 1 (and other similar figures)
It’s not clear to me what is being plotted on the x and y axes.
Answer: This is a common presentation of OPLS-DA score plots. The x-axis is the result of a linear combination of the original variables (lipids) and shows the predictive component. The y-axis shows the orthogonal component.